# Perceptions, experiences, and beliefs regarding urinary tract infections in patients with neurogenic bladder: A qualitative study

**Margaret A. Fitzpatrick**[1,2]*, **Pooja Solanki**[3], **Marissa Wirth**[3], **Frances M. Weaver**[3,4], **Katie J. Suda**[5,6], **Stephen P. Burns**[7,8], **Nasia Safdar**[9,10], **Eileen Collins**[11], **Charlesnika T. Evans**[3,12]

1 Center of Innovation for Veteran-Centered and Value-Driven Care, Rocky Mountain Regional VA Medical Center, Aurora, CO, United States of America, 2 Department of Medicine, Division of Infectious Diseases, University of Colorado Anschutz Medical Campus, Aurora, CO, United States of America, 3 Center of Innovation for Complex Chronic Healthcare, Edward Hines, Jr. VA Hospital, Hines, IL, United States of America, 4 Parkinson School of Health Sciences and Public Health, Loyola University Chicago, Maywood, IL, United States of America, 5 Center for Health Equity Research and Promotion, VA Pittsburgh Healthcare System, Pittsburgh, PA, United States of America, 6 Department of Medicine, Center for Research on Healthcare, University of Pittsburgh School of Medicine, Pittsburgh, PA, United States of America, 7 Spinal Cord Injury/Disorders Service, VA Puget Sound Healthcare System, Seattle, WA, United States of America, 8 Department of Rehabilitation Medicine, University of Washington School of Medicine, Seattle, WA, United States of America, 9 Department of Medicine, Division of Infectious Diseases, University of Wisconsin-Madison School of Medicine and Public Health, Madison, WI, United States of America, 10 Department of Medicine, Division of Infectious Diseases, William S. Middleton VA Hospital, Madison, WI, United States of America, 11 College of Nursing, University of Illinois Chicago, Chicago, IL, United States of America, 12 Center for Health Services and Outcomes Research, Northwestern University Feinberg School of Medicine, Chicago, IL, United States of America

* margaret.fitzpatrick@va.gov

**Data Availability Statement:** Data cannot be shared publicly because the United States Department of Veterans Affairs (VA) places legal

## Abstract

Although urinary tract infections (UTIs) are common in patients with neurogenic bladder (NB), limited data exist on UTI perceptions, experiences, and beliefs in these patients. We recruited adults with NB due to spinal cord injury/disorder (SCI/D) or multiple sclerosis (MS) at three Veterans Affairs (VA) medical centers to participate in 11 virtual focus groups. Audio transcripts were coded using a mixed approach with primary deductive codes linked to the Health Belief Model, and secondary inductive codes informed by grounded theory. Twenty-three Veterans (SCI/D, 78%; MS, 18.5%) participated between May 2021 and May 2022. Participants' perspectives, experiences, and beliefs about UTI were reflected in three major themes: 1) influence of caregivers; 2) influence of the healthcare environment and provider characteristics; and 3) barriers and facilitators to care. Caregivers promoted care-seeking behavior, enabled in-home care, and enhanced participants' self-efficacy to understand educational material. Participants had poor perceptions of providers who were not knowledgeable about NB or ineffectively communicated. Good relationships with providers who knew the participant well improved self-efficacy to follow provider recommendations. These results suggest that patient-centered interventions to improve UTI management in this population should expand caregiver involvement, enhance patient-provider communication, and target provider types and care settings that lack familiarity with NB.

restrictions on access to Veteran's health care data, which includes both identifying data and sensitive patient information. The minimum dataset used for this study is located on VA computer servers behind a firewall. The dataset is available to researchers with an approved VA study protocol which meets the criteria for access to confidential Veteran data. For more information on access to VA data, please visit https://www.virec.research.va. gov or contact the VA Information Resource Center (VIReC) at virec@va.gov.

**Funding:** This work was supported by the U.S. Department of Veterans Affairs, Veterans Health Administration, Office of Research and Development (https://www.research.va.gov/) Rehabilitation Research and Development Career Development Award (B2826-W to M.A.F.) and Research Career Scientist Award (RCS 20-192 to C.T.E.; RCS 98-35 to F.M.W ). The funders had no role in study design, data collection and analysis, decision to publish, or preparation of the manuscript.

**Competing interests:** The authors have declared that no competing interests exist.

## Introduction

Many patients with chronic neurologic injuries and disorders such as spinal cord injury and disorder (SCI/D) and multiple sclerosis (MS) experience chronic bladder dysfunction termed neurogenic bladder (NB) [1]. Urinary tract infections (UTIs) are one of the most frequent complications in patients with NB. An estimated 34% of patients with NB have at least one UTI diagnosis each year, and patients with SCI/D have an average of 2.5 UTI diagnoses/year [2, 3]. Urinary tract infections in patients with NB often lead to hospitalization, and they can be associated with adverse clinical outcomes such as kidney stones, renal failure, sepsis, and even death [4–6].

Patients with NB have a high prevalence of chronic bladder colonization with bacteria in the absence of true infection (termed 'asymptomatic bacteriuria' or ASB), particularly in the setting of chronic indwelling catheter use [7–9]. Routine antibiotic treatment of ASB is not recommended due to lack of clinical benefit, whereas antibiotic treatment of UTI prevents many adverse outcomes [10, 11]. Prior studies have shown that UTIs are often misdiagnosed and inappropriately treated in patients with NB, which may relate in part to the high frequency of non-specific symptoms in this population [12]. This can lead to antibiotic overuse, which can cause bacterial resistance, adverse drug reactions, and diarrheal illness from *Clostridioides difficile* infection—outcomes that ultimately prolong healthcare lengths of stay and increase costs, morbidity, and mortality [13–15].

Although we have data on the medical impact of UTIs and clinical outcomes, qualitative studies exploring the individual UTI experience are lacking. Most prior studies have examined *healthcare provider* knowledge, attitudes, and beliefs about UTI within the focused context of inappropriate treatment and antibiotic overuse [16–19]. Comparatively little work has been done to assess *patient* perspectives, experiences, and beliefs about UTI, with most prior studies again focusing on treatment and antibiotic use in general populations rather than patients with NB [20–22]. The few prior studies assessing UTI experiences among patients with NB have identified diverse themes related to ambiguous symptoms, stigma, care-seeking behavior, the patient-provider relationship, and antibiotic use [23–26]. In this study, our objective was to characterize perspectives, experiences, and beliefs about UTI among patients with NB. Results from this study can improve patient-centered UTI care by giving healthcare providers and researchers a more comprehensive understanding and appreciation of patients' individual health experiences and perspectives and how they shape behavior.

## Methods

This analysis represents part of the qualitative research conducted in a larger mixed-methods project that aims to inform the development of a patient-centered intervention to improve clinical and patient-reported outcomes related to ASB and UTI in patients with NB. We recruited adult Veterans from three Veterans Affairs Medical Centers (VAMCs) with NB and underlying SCI/D or MS to participate in virtual focus group discussions conducted via the Microsoft Teams platform. Focus groups were chosen because they emphasize interaction among participants. The ability of participants to ask questions, exchange perspectives, and comment on each other's points of view in a focus group is particularly useful for exploring beliefs and experiences, which was the primary objective of this study. Focus groups also offer the advantage of gathering data from multiple Veterans with NB at one time, potentially providing a more comprehensive and robust assessment across this diverse patient population. The Institutional Review Board at the Edward Hines, Jr. VA Hospital approved this study. The

methods and results from this study are reported in accordance with guidance from the Consolidated Criteria for Reporting Qualitative Research (COREQ) checklist [27].

## Research team

All focus group discussions were led by the Principal Investigator (PI; MF), an Infectious Diseases (ID) physician with training in Health Services Research (HSR). The PI did not have a pre-existing relationship with or provide clinical care to study participants. Two team members with training in Public Health and HSR assisted with data collection, coding, analysis, and interpretation (MW, PS). There was also a diversity of perspectives and training among the broader research team, which included MD, PhD, and PharmD researchers with experience in ID, Rehabilitation Medicine, HSR, and Implementation Science.

## Study setting and sample

Focus group participants were Veterans with NB due to SCI/D or MS who had in-person encounters with a UTI diagnosis between 2017–2018 at three VAMC sites. Two sites have VA SCI/D Centers ('Hubs') that are referral centers for large geographic areas and provide acute SCI/D rehabilitation and life-long follow-up care, while the third is a 'Spoke' site where healthcare providers work closely with coordinating SCI/D Hub facilities to provide comprehensive primary and specialized care. All three sites are affiliated with the national VA MS care network (MS Centers of Excellence). Veterans with at least one UTI encounter in the year prior to the focus group were eligible. Diagnoses of SCI/D, MS, and UTI were defined primarily by International Classification of Diseases version 10 (ICD-10) codes [12]. We aimed to include a mix of patients with both low and high UTI frequency, which was defined based on the number of UTI encounters for each patient that occurred between 2017–2018 (low = 1 to 4 encounters; high = > 4 encounters). Twenty-three Veterans participated in 11 virtual focus group discussions. See Fig 1 for a description of study recruitment. Caregivers of eligible patients were not specifically recruited or instructed to participate in focus group discussions; however, their participation was not prohibited. A total of four non-medically trained caregivers (e.g., spouse) ended up participating in three different focus groups.

## Data collection

Participants were recruited and focus group discussions were conducted via Microsoft Teams between May 2021-May 2022. Discussions lasted between 1–2 hours. The conceptual model underlying the design and analysis of this study was the Health Belief Model (HBM) [28]. The HBM identifies constructs that help explain health-related behaviors and is one of the most widely established and utilized behavior theories in health research (Fig 2). The HBM centers on identifying people's beliefs about illnesses and their consequences ('individual beliefs') which can impact health-related behaviors and actions ('actions'). The individual beliefs encompass perceived risks and severity of illness, benefits of and barriers to health-promoting actions, and perceived self-efficacy to benefit from health-promoting actions. Various modifying factors and external cues (e.g., advice from the internet, healthcare providers) can also influence individual beliefs (Fig 2). A multidisciplinary team including physicians, pharmacists, and researchers helped develop a focus group discussion guide (S1 Fig). The guide was informed by the HBM, whereby we mapped question prompts about ASB and UTI beliefs and experiences to individual beliefs from the HBM. Question prompts and follow-up probes in the discussion guide were iteratively modified throughout the study as new concepts and ideas emerged and in response to questions or probes that were misunderstood or confusing to the initial focus group participants. Data were collected between May 2021 and May 2022 via

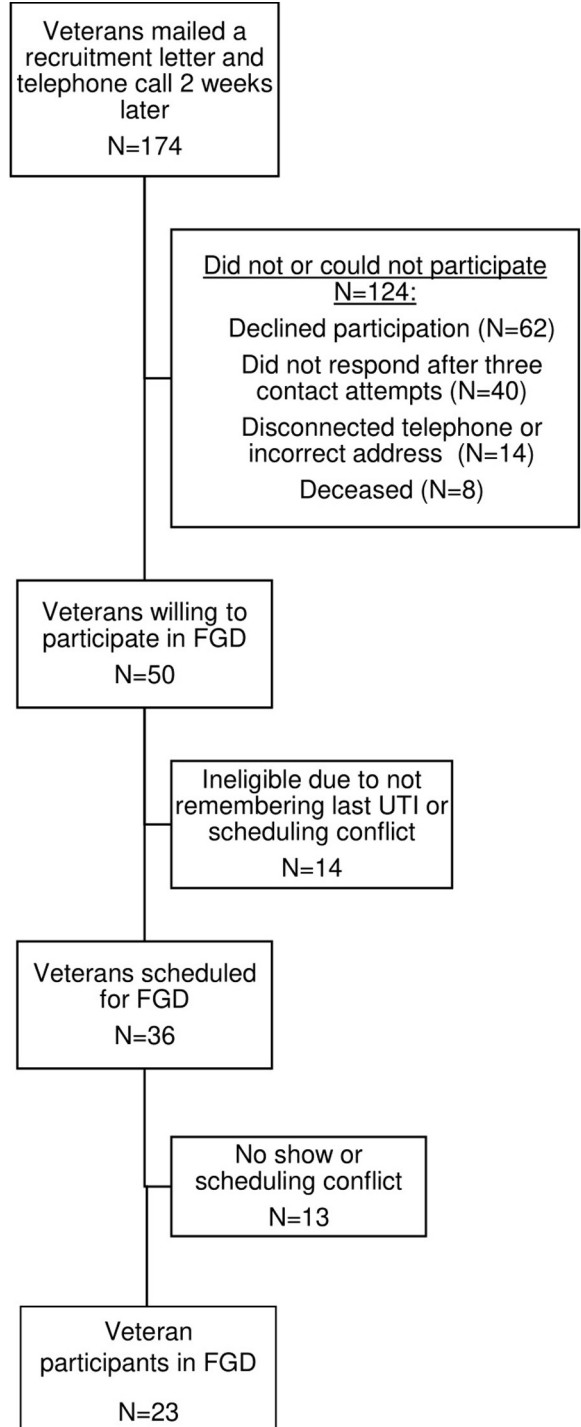

**Fig 1. Flowchart of study recruitment and participant identification.** Flowchart depicting numbers of patients who were contacted, recruited, scheduled, and participated in focus group discussions. UTI, urinary tract infection; FGD, focus group discussion.

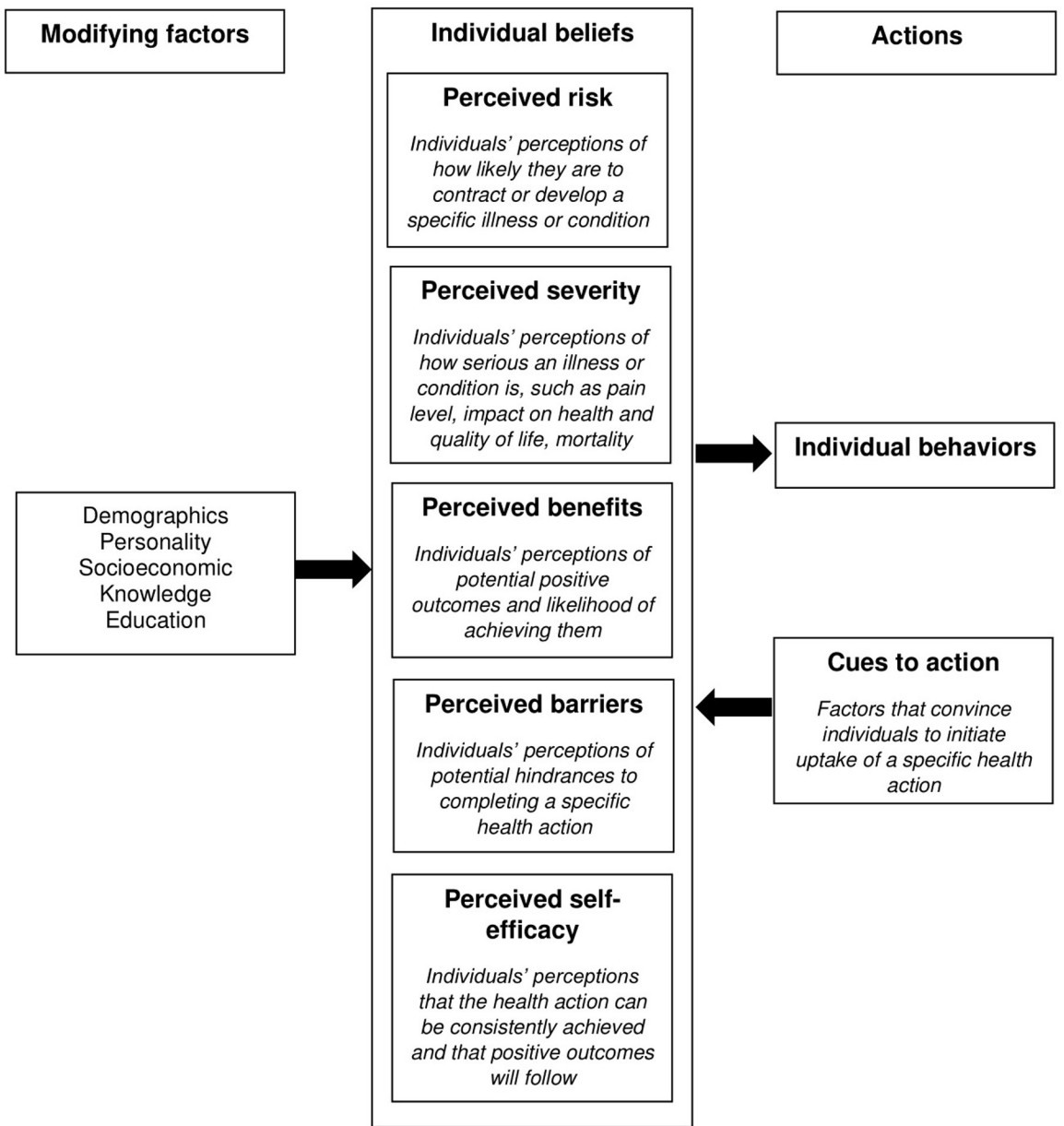

**Fig 2. Diagram of the Health Belief Model (HBM).** Major constructs that comprise the HBM conceptual framework for individual health-related behaviors.

audio recordings and transcriptions of the virtual focus groups. As part of the recruitment process, four authors (MF, MW, PS, CE) had access to identifying information about individual participants during and after data collection.

## Data analysis

Data were accessed for this analysis between May 2021 –May 2023, which encompasses the period of data collection, data analysis, and dissemination. We took a mixed approach to thematic analysis of transcripts and field notes, using deductive analysis by applying theory to the data and inductive analysis by allowing themes and codes to emerge as the data were reviewed.

Three study team members (MF, PS, MW) independently reviewed and coded all data. An initial codebook with deductive codes was developed a priori based on HBM constructs and key topics of predetermined interest. During analysis, inductive in vivo codes were created to capture the products of group discussion and unanticipated topics.

Throughout the analytic process, team members refined codes iteratively and with consensus through serial review of transcripts and identification of relationships between emerging themes. Codes were synthesized to identify broader themes using memo writing and tabling techniques. We aimed to conduct a minimum of six focus groups (two per study site) with 4–6 participants per group. Due to lower than anticipated numbers of participants per group in the first few focus groups and difficulty with recruitment at one of our sites, we added additional focus groups to ensure we captured a diverse range of concepts and themes. We ultimately conducted 11 focus groups, at which point we observed no more new emergent themes. Saturation was confirmed using the method reported by Guest et al., with a new information threshold set at $\leq 0.5\%$ [29]. Nvivo software (version 12) was used to facilitate coding and data synthesis.

## Results

Participant characteristics are detailed in Table 1. Most participants were male (96%), older (mean age 65), had underlying SCI/D (78%), and had low UTI frequency (78%). All participants used some form of catheterization for bladder management. Three major themes regarding patient perceptions, experiences, and beliefs about UTIs emerged: 1) the influence of

**Table 1. Characteristics of focus group participants.**

| Characteristics | Participants (n = 23) |
|---|:---:|
| Age, mean (SD) | 65 (11) |
| Gender | |
| Male | 22 (96) |
| Female | 1 (4) |
| Race | |
| White | 18 (78) |
| Black or African American | 4 (17) |
| Native Hawaiian/Pacific Islander | 1 (4) |
| Ethnicity | |
| Non-Hispanic or Latino | 21 (91) |
| Hispanic or Latino | 0 |
| Missing | 2 (9) |
| Neurologic injury or disease | |
| SCI/D | 18 (78) |
| MS | 5 (22) |
| Bladder management | |
| External (condom) catheter | 3 (13) |
| Intermittent catheterization | 7 (30) |
| Indwelling catheter | 13 (57) |
| UTI frequency | |
| Low (1–4 encounters) | 18 (78) |
| High (>4 encounters) | 5 (22) |

SCI/D, spinal cord injury/disorder; MS, multiple sclerosis

caregivers; 2) the influence of the healthcare environment and provider characteristics; and 3) barriers and facilitators to care. These themes primarily relate to HBM constructs as detailed in Fig 3.

## Theme 1: Caregivers influence UTI perceptions, experiences, and beliefs

Most participants described experiences with caregivers, either medically trained in-home healthcare providers (e.g., a visiting nurse), non-medically trained in-home caregivers (e.g., a family member), or both. For these participants, UTI perceptions, experiences, and beliefs were often influenced by caregiver interactions. Participants described that caregivers were familiar with them and their needs (Table 2, quotes A, D) and they often coordinated UTI prevention, diagnosis, and treatment activities. Overall, participants perceived mostly beneficial UTI experiences with caregivers, particularly medically trained in-home healthcare providers (Table 2, quotes A-C). A few participants noted negative experiences with caregivers, which centered around nurses not listening to the participant's concerns about catheter hygiene or not arriving on time to administer medications.

**Caregivers enable and coordinate UTI care.** Many participants said that caregivers enabled faster, easier, and more coordinated access to UTI care. Visiting nurses allowed participants to remain at home for UTI-related care such as catheter changes and urine sample collection (Table 2, quote B), rather than traveling to a clinic or hospital, which was noted to be particularly burdensome by some participants when experiencing UTI symptoms. One participant even likened his visiting home nurse to an 'ambassador' between himself and other healthcare providers (Table 2, quote C).

**Caregivers help participants recognize and interpret UTI symptoms and decide when to seek care.** Participants perceived caregivers as helpful in recognizing and interpreting UTI symptoms, particularly when symptoms may have impaired the participant's own cognitive abilities. Caregivers often engaged in shared decision-making about seeking care. Caregiver assessments about whether participants' symptoms were concerning for UTI and warranted further action affected the participants' perceptions and beliefs of whether they had a UTI (Table 2, quotes D-F). A good example is a patient's wife who participated in FG 13 noting that changes in his urine output, color, consistency, and sediment prompted her concern for UTI (Table 2, quote F).

**Caregivers help participants understand education and information about UTI.** Some participants perceived that caregivers helped them better understand information received from healthcare providers about UTI. They preferred that caregivers be present when they were receiving education or information about UTI, with one participant indicating that healthcare providers should be more willing to interact with his wife without him being present (Table 2, quote G). In this context, caregivers may facilitate participants' self-efficacy by helping them follow provider recommendations and process information about UTI.

## Theme 2: The healthcare environment and provider characteristics influence UTI perceptions, experiences, and beliefs

Many participants described ways in which their perceptions, experiences, and beliefs about UTIs were influenced by the healthcare environment and provider characteristics, including the provider specialty type, the care setting, and aspects of patient-provider communication. Participants most frequently interacted with primary care providers (PCP) for long-term UTI care, including prevention. PCPs were most often a physician or physician's assistant, with some participants indicating that their PCP had specific training in SCI/D. In contrast, most participants who experienced acute UTI symptoms described receiving diagnosis and

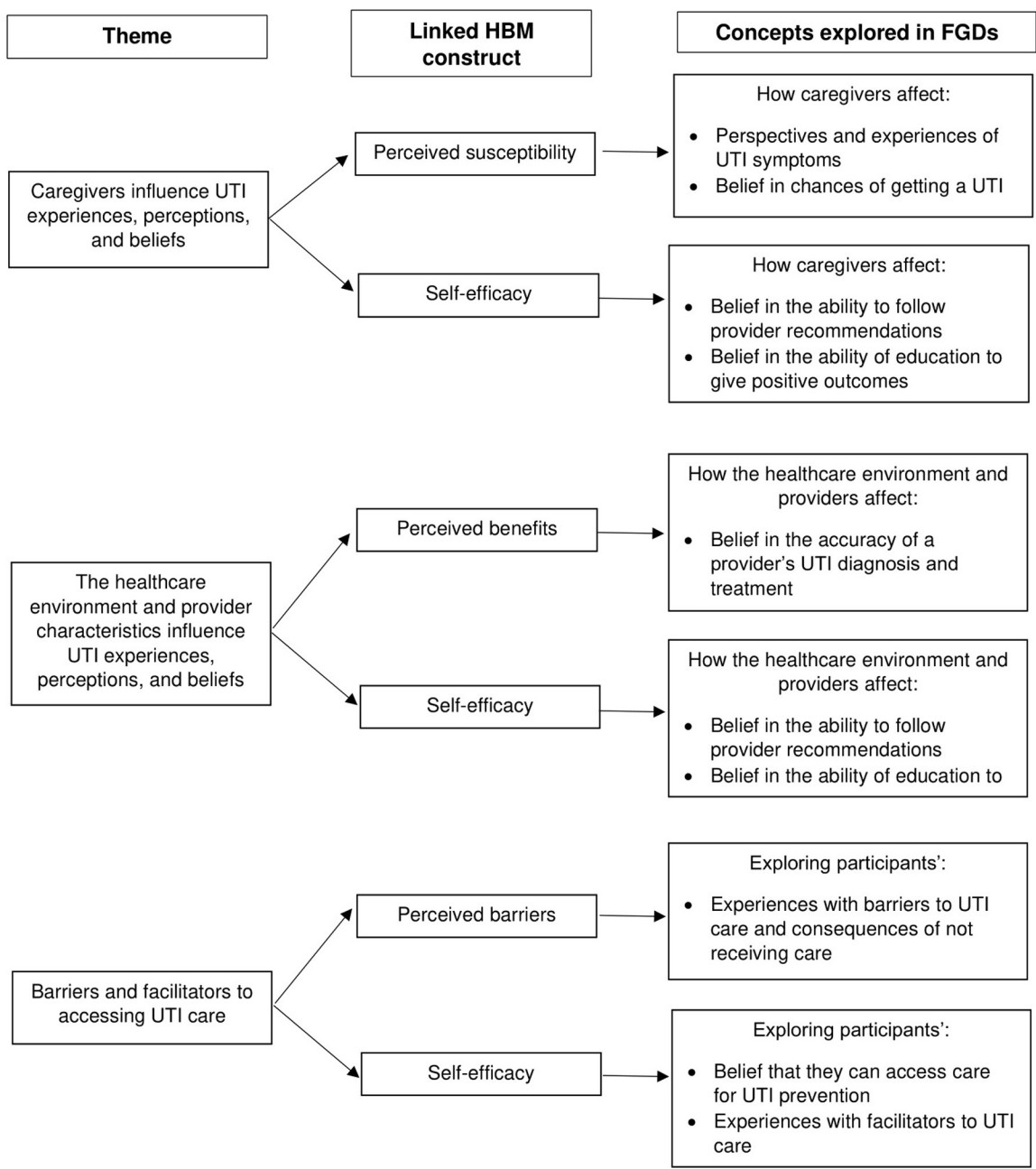

**Fig 3. Major themes with links to Health Belief Model.** Three major themes identified in this study and their links to the HBM constructs and other concept(s) derived from focus group discussions. HBM, Health Belief Model; FGD, focus group discussions; UTI, urinary tract infection.

treatment in the Emergency Department (ED), with only a minority describing care in other locations such as their PCP's clinic, urgent care, or VA home-based primary care. The speed and severity of symptom onset often necessitated ED care rather than other outpatient locations (Table 3, quote A).

**Provider and care setting characteristics affect perceptions, experiences, and beliefs about the quality of UTI care.** Participants perceived greater benefits of UTI care from providers who were knowledgeable about and experienced with NB, with this perception

Table 2. Participant quotes supporting Theme 1 and its associated categories.

| Theme | Category | Quotes |
|---|---|---|
| Caregivers influence UTI experiences, perceptions, and beliefs | Caregivers are familiar with participant needs | A. "I have caregivers all week, so I'm fortunate there and I get help with that. One of them is an LPN so she's very in tune to everything that goes wrong with me. . .It's good having an outside person looking in, because when they know you, they see when you're off. They know more when you're off." (FG7) |
| | Caregivers enable and coordinate UTI care | B. "Take one of my wonderful nurses right here. So if I feel like I got an infection, the first thing I would do is say: Hey, I think I got a bladder infection 'cause I'm hurt down there. I feel like my bladder is being squeezed. . . nine times out of 10, they'll say: Okay, we'll see if we can get you a test. . . They'll come in and change my catheter, and change the bag, and then when the new bag gets some urine in it, they'll check the urine." (FG5) C. ". . . a nurse that comes to your home to look over your situation and she'll explain things. . .She actually acts as your ambassador, she'll look at your situation and she'll say, oh you need this, and she will go back and talk to the doctor and the *[physical therapy]* person and all that. Because when this all happens you have no idea what, what's going on." (FG13) |
| | Help participants recognize and interpret symptoms and decide when to seek care | D. "And then this last one I was getting these fevers. I was feelin' really strange. And my nurse just happened to come over for my monthly visit. She thought I had an aneurysm, 'cause that, I wasn't myself. She knows how I usually am. I go let me, I'll go to the emergency room at the VA and that's what I did." (FG12) E. "I got aides and nurses that come by my house . . . So I can ask one of them: Hey, I feel like I'm getting a bladder infection. What do you think I need to do?" (FG 5) F. ". . .when he's not producing as much urine then it's always suspicious that it might be a bladder infection. The urine is usually darker under 200 [mLs]. . .when I cath him I let him know how much and what color and if it's cloudy at all we'll even hold it to the light to see if there are particles floating in it or if it's just that it's cloudy urine." (FG13, Caregiver) |
| | Help participants understand education and information about UTI | G. "I think that the brochure is always helpful. . . a little information in my head helped me understand what was happening. . . But my wife needs to know. . .. I think they should be more willing to talk to our spouse about what's going on with me—and maybe they should talk to them privately about what's going on with me, so that I don't interfere with them telling her, because I think she needs to know." (FG3) |

sometimes affecting behavior regarding UTIs. For example, one participant preferred waiting longer or traveling farther to go to a clinic or facility where the staff and providers know him well (Table 3, quote B). Several participants also perceived better experiences with UTI care at the VA, particularly the VA SCI/D unit (Table 3, quote C). In contrast, one participant

**Table 3. Participant quotes supporting Theme 2) and categories.**

| Theme | Category | Quotes |
|---|---|---|
| The healthcare environment and provider characteristics influence UTI experiences, perceptions, and beliefs | Provider and care setting characteristics affect perceptions, experiences, and beliefs about the quality of UTI care | A. "I'd rather be able to go just to the SCI clinic, obviously, but I don't know until it's too late, so I end up going to the emergency and the only drawback to going to the emergency is you don't end up in the Spinal Cord Injury unit." (FG8)<br>B. "I don't think I get better care anywhere else... I know my primary and my primary knows me. All the other people on the staff know what my issues are. ...I'd rather wait out a weekend and —rather than going to an ER in a foreign hospital, because it's gonna take three or four days for them with the learning curve to catch up anyway, than if I go ahead and get to [VA site], where I know we know what's going on . . ." (FG6)<br>C. "I like going [to the VA] because they know me... A lot of the nursing staff have done tours in SCI and have gone into some of the general wards, .and so you always run into somebody that you knew or you used to know from down there in SCI, and so the treatment there is better." (FG11) |
| | Negative perceptions and experiences of patient-provider communication regarding UTIs | D. "One word. Listen. They don't know our bodies, they don't know our minds, just 'cause we can't put it in medical terms, terminology, we still can tell them how we feel if they just listen. Okay?" (FG3)<br>E. "The doctors should go to a class: Listen to your patient... because we know, because we're going through it. They're not feeling our pain or our symptoms." (FG7)<br>F. "I want them to listen, and to know that everybody is not textbook classic and if they look, review my records, they'll see that when I'm ignored, a lot of times I end up in intensive care which is really aggravating and could have been avoided. And it's upsetting because a lot of times you feel like nobody is listening to what you say." (FG4)<br>G. "I've had various antibiotics. . .a couple of them made me sick to my stomach because they were so strong. And that's about all they tell me. But generally, they don't tell me what it is. I have to read it, you know, once I get the prescription." (FG7) |

described better experiences for UTI care with his PCP compared to a SCI/D specialist because he has an established, trusting relationship with his PCP. Finally, several participants described negative experiences with care received in the ED from providers who were not knowledgeable about NB or about the participant's medical history. One participant said:

"...I don't look forward to when I go, if I'm going into the emergency room. ... they're aware of people having spinal issues, but they don't know what to do about it... my wife had to come down early in the morning and take care of the bowel program for me, and that kind of behavior is just lacking in the emergency room. Everybody walks in there, but not many people roll in there, and so I think that's where we really feel shortchanged." (FG8)

**Negative perceptions and experiences of patient-provider communication regarding UTIs.** Many participants and caregivers described UTI encounters where they felt the providers did not listen to or address their concerns (Table 3, quotes D-F). This perception led some participants and caregivers to feel a disconnect or conflict between what they told providers and what providers diagnosed or recommended (e.g., a participant suspected UTI and the provider did not). One participant described a UTI encounter:

**Table 4. Participant quotes supporting Theme 3) and categories.**

| Theme | Categories | Quotes |
|---|---|---|
| Barriers and facilitators to accessing UTI care impact UTI perceptions, experiences, and beliefs | Barriers to accessing UTI care | A. ". . . because of my lack of mobility, it's not easy. I live on the second floor of an apartment, with stairs, winding stairs, so it's not easy for me, even when I'm healthy, to get in and out. . . I need to have the assistance of an ambulance to get me out of the house, because my legs are so weak . . ." (FG3)<br>B. "I've been going to a local doctor now for years, because it was more convenient. . . as soon as they can identify it and give an antibiotic, I'll be back on track. . . I know going to the VA can save me a lot of money on a lot of things, but for that, it was just quicker and expedient. I want to get it handled right this minute." (FG3)<br>C. ". . .where I am, you really can't go right to the VA hospital, because it's too far. They have to put you at a community hospital, and they don't have all my records. So that's a really frustrating aspect of the UTIs, because I always end up going to [hospital] in [city]." (FG7)<br>D. "We stopped going to the VA because their protocol is ridiculous. If I waited until he had everything they wanted—we'd be burying him. So we just go to the local [urgent care] clinic here. . . I don't have to wait for a full-on fever. I don't have to wait for this or dark urine or smelly urine, or all —I mean, I'm appreciative that they do have a protocol, but [the city the VA is in] is two hours away. . .. So we just ended up going to the local clinic. . ." (FG7)<br>E. ". . .Self—self-administer hysteria. Nah, I did absolutely nothing for—until the second day. Which is probably way too long." (FG3) |
| | Facilitators to accessing UTI care | F. "Well, my first call usually is to the spinal cord injury unit. . . And if I need a local examination, there's a VA nurse through the Home-Based Primary Care program the VA runs. They kind of work in conjunction with the spinal cord injury unit up in [city]. . ." (FG11)<br>G. "I have a nurse that comes out every month and checks me out. . .if I feel like I'm getting on the verge or I think I might be starting, I'll give him a urine sample right there, he'll have it tested immediately, and if there is the start of a UTI, the doctor that's part of the [Home Based Primary Care] program will issue the antibiotics and my wife will just run out to the VA. . .. So that's one of the advantages I have. . . Unless it's really severe and sudden. . . I can have my nurse come by. . .and then he'll take care of it from there." (FG 4)<br>H. ". . . when they discovered that he's susceptible to autonomic dysreflexia, it was because the catheter he had in the hospital had been crimped and the bladder wasn't emptying. . . I had never heard of such a thing. But they trained me on how to help with the cathing and I became basically his bowel and bladder person." (FG11, caregiver)<br>I. "I always have my wife with me. She knows what drugs I'm on. I know some of them, but I don't know all of them, and she can easily tell the doctor and nurse what I'm on and when I take it." (FG4) |

"And that's when the doctors say: we don't really think you have a urinary tract infection, because you don't have a fever. . . I feel like I'm sitting on a stick. . . I just have a feeling that I don't feel good . . . And when I would mention that to the doctors, they would go: You don't have a temperature. We'll do a urinalysis really quick. Well, you don't have enough bacteria in your urine for UTI. So we're not gonna treat it. . . And it was like, swept under the rug, and I ended up in intensive care both times." (FG 4)

Negative experiences with poor patient-provider communication were often associated with negative emotions such as anger, frustration, and irritation (Table 3, quote F). Finally, some participants also described that providers did not answer questions or communicate important information about UTI prevention, diagnosis and treatment, such as possible antibiotic side effects (Table 3, quote G).

### Theme 3: Barriers and facilitators to accessing care impact UTI perceptions, experiences, and beliefs

Participants experienced various barriers to accessing care when they suspected a UTI, many of which related to physical and functional limitations of their underlying neurologic injury or disorder. One participant described difficulties in leaving his house to see a provider for suspected UTI, indicating he often requires an ambulance to transport him (Table 4, quote A). Although many participants perceived better care at the VA compared to non-VA facilities, limited mobility and difficulty with transportation made it more convenient and easier for some participants to go to local clinics (Table 4, quotes B, C). In contrast, one participant and his caregiver perceived that specific requirements and protocols of a local VA clinic were a barrier to receiving timely UTI care, and, as a result, they often sought care at a non-VA urgent care clinic (Table 4, quote D). A few participants related that their own thoughts and emotions were barriers to seeking care (Table 4, quote E). Perceived barriers were often manifested as delays in getting care, the consequences of which led to participant anxiety and concern about infection progressing and becoming more severe.

Several participants described aspects of the VA healthcare system that helped facilitate access to care. The VA's Home-Based Primary Care program and visiting nurses were perceived as facilitators to care since they allowed the participant to remain at home, thereby overcoming mobility and transportation barriers (Table 4, quotes F, G). This perceived benefit was described mainly in relation to UTI diagnosis and treatment (e.g., urine sample collection, care coordination for treatment; Table 2, quote B and Table 4, quotes F, G). Non-medically trained caregivers also facilitated UTI prevention care for participants by managing catheter maintenance and hygiene (Table 4, quote H) and prescription medications (Table 4, quote I).

### Discussion

Although variability exists in perceptions of and beliefs about UTI among patients with NB, several themes are common to many patient experiences. Interpreting these themes within the HBM constructs allowed us to explore how these perceptions, experiences, and beliefs influence patient behavior, such as care-seeking decisions, expectations for the encounter, and ability to follow provider recommendations. Most existing data on patient UTI perspectives, experiences, and beliefs are from studies of women with recurrent UTI. In a recent systematic review and meta-ethnography on patient UTI experiences, only three studies included patients with NB, and all of them were conducted outside the U.S [23]. Our study provides important data characterizing the complexity of the UTI experience among patients with NB and can help tailor patient-centered interventions to improve UTI management in this population.

Our first major theme involved the influence of caregivers, which we described both for non-medically trained family caregivers and in-home visiting nurses. Unfortunately, studies of caregiver experiences for patients with NB remain limited. One prior study focused on caregivers of children with NB and, therefore, has limited application to our patient population; however, that study had similar results to ours whereby caregivers felt that healthcare providers in the community (particularly non-specialist providers) lacked knowledge about UTI, which led to patients and caregivers not feeling listened to or believed [30]. Our findings suggest that involving caregivers in comprehensive UTI management and developing education specifically for caregivers that encompasses symptom assessment, shared medical decision-making, and creation of prevention and treatment plans will likely improve the patient experience and may enhance adherence to provider recommendations for care.

Our second major theme involved the influence of the healthcare environment and provider characteristics. Poor communication with providers, feeling ignored, and perceiving that symptoms or concerns were discounted was common, often with providers who were not knowledgeable about NB (such as in the ED). This was a frustrating and distressing aspect of the patient UTI experience. Similar findings have been observed in studies of women with recurrent UTI, where patients perceived that providers lacked knowledge and awareness of UTI, ignored or underestimated UTI burden, and trivialized the disease experience [22, 31, 32]. This led to perceptions of inadequate care and beliefs that they were not getting the correct treatment and diagnosis, which is similar to our findings of patients perceiving a 'disconnect' between what they told providers and what providers ultimately diagnosed. Likewise, 'being heard, seen, and cared for with dignity' was one of the major themes identified in the systematic review of patient UTI experiences [23]. Poor patient-provider communication and resulting negative emotions could lead patients to perceive fewer benefits from following a provider's diagnosis and treatment recommendations, particularly regarding antibiotic treatment. It could also diminish self-efficacy by lowering patients' beliefs in their ability to follow provider recommendations and benefit from education. In our study, positive perceptions and experiences of UTI care occurred when participants trusted and were familiar with their provider and care setting. Various participants described this occurring with their primary care provider, visiting home nurse, Physical Medicine & Rehabilitation specialist, or with care received on the SCI/D unit. Prior studies of UTI experiences in intermittent catheter users and women with recurrent UTI also demonstrated that more patients had positive perceptions of UTI care with providers who knew them well, listened to them, engaged in shared medical decision-making, and approached care on an individual basis [26, 33].

Our final theme involved barriers and facilitators to accessing UTI care, many of which specifically related to physical and functional limitations due to the underlying chronic neurologic injury or disorder. Similar to our results, where greater distance from a VA facility was a barrier to care, geographic distance from specialists and subsequent delays in treatment were noted as two barriers to effective UTI care in a large survey study of women with recurrent UTI [22]. Overall, few prior studies have described patient perceptions and experiences of barriers and facilitators to UTI care; thus, our study provides valuable and novel data that could assist providers and larger healthcare systems with developing strategies to overcome barriers and support facilitators to UTI care.

Our study has several important limitations. First, it was conducted in the VA and only included Veterans, with most participants being male and older. The VA cares for one of the largest populations of patients with chronic neurologic injuries and disorders in the U.S.; however, our results may not fully represent perceptions, experiences, and beliefs of non-Veteran populations with NB, particularly women. Second, although we chose focus group discussions because participants could ask questions, exchange perspectives, and comment on each other's

experiences, semi-structured interviews may have allowed more in-depth exploration of these concepts. Third, all focus group discussions were conducted virtually through the Microsoft Teams platform and our study population was limited to participants who could access Microsoft Teams. To help overcome this, study team members created guides to facilitate access using a variety of devices, including laptops, desktops, and mobile phones.

In conclusion, UTI perceptions, experiences and beliefs were significantly influenced by caregivers, the healthcare environment, and patient-provider interactions. Caregiver knowledge and involvement along with established and trusted provider relationships facilitates perceptions of high-quality UTI care in persons with NB, while greater distance to care and settings where providers are not familiar with NB in disabled populations (such as the Emergency Department) are barriers. Interventions to enhance patient-centeredness of UTI care for these individuals could focus on expanding in-home care by trusted, knowledgeable providers and improving patient-provider communication.

## Supporting information

**S1 Fig. Sample focus group discussion guide document.**
(PDF)

## Author Contributions

**Conceptualization:** Margaret A. Fitzpatrick, Frances M. Weaver, Katie J. Suda, Stephen P. Burns, Nasia Safdar, Eileen Collins, Charlesnika T. Evans.

**Data curation:** Margaret A. Fitzpatrick, Pooja Solanki, Marissa Wirth.

**Formal analysis:** Margaret A. Fitzpatrick, Pooja Solanki, Marissa Wirth, Frances M. Weaver, Charlesnika T. Evans.

**Funding acquisition:** Margaret A. Fitzpatrick, Frances M. Weaver, Katie J. Suda, Stephen P. Burns, Nasia Safdar, Eileen Collins, Charlesnika T. Evans.

**Investigation:** Margaret A. Fitzpatrick, Marissa Wirth, Katie J. Suda, Stephen P. Burns, Nasia Safdar, Charlesnika T. Evans.

**Methodology:** Margaret A. Fitzpatrick, Pooja Solanki, Marissa Wirth, Frances M. Weaver, Katie J. Suda, Stephen P. Burns, Nasia Safdar, Eileen Collins, Charlesnika T. Evans.

**Project administration:** Margaret A. Fitzpatrick, Marissa Wirth.

**Supervision:** Margaret A. Fitzpatrick, Charlesnika T. Evans.

**Validation:** Margaret A. Fitzpatrick.

**Writing – original draft:** Margaret A. Fitzpatrick, Pooja Solanki, Marissa Wirth.

**Writing – review & editing:** Margaret A. Fitzpatrick, Pooja Solanki, Marissa Wirth, Frances M. Weaver, Katie J. Suda, Stephen P. Burns, Nasia Safdar, Eileen Collins, Charlesnika T. Evans.

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
