## [Decision Letter · Decision Letter 0]

12 Sep 2023

PONE-D-23-15723Perceptions, experiences, and beliefs regarding urinary tract infections in patients with neurogenic bladder: a qualitative studyPLOS ONE

Dear Dr. Fitzpatrick,

Thank you for submitting your manuscript to PLOS ONE. After careful consideration, we feel that it has merit but does not fully meet PLOS ONE’s publication criteria as it currently stands. Therefore, we invite you to submit a revised version of the manuscript that addresses the points raised during the review process.

Thank-you for your patience with this submission. As you can see both reviewers are asking for just a little more detail to be added in the introduction, methods, themes and discussion. I think they have given helpful advice to strengthen the paper. 

We look forward to receiving your revised manuscript.

Kind regards,

Maria Berghs, PhD

Academic Editor

PLOS ONE

Reviewers' comments:

Reviewer's Responses to Questions

**Comments to the Author**

1. Is the manuscript technically sound, and do the data support the conclusions?

Reviewer #1: Yes

Reviewer #2: Yes

2. Has the statistical analysis been performed appropriately and rigorously? 

Reviewer #1: Yes

Reviewer #2: N/A

3. Have the authors made all data underlying the findings in their manuscript fully available?

Reviewer #1: Yes

Reviewer #2: Yes

4. Is the manuscript presented in an intelligible fashion and written in standard English?

Reviewer #1: Yes

Reviewer #2: Yes

5. Review Comments to the Author

Reviewer #1: This is a qualitative analysis of patients perceptions of UTI in veterans with neurogenic bladder. This is a difficult patient population to treat. The team did a series of focus groups to understanding patients perceptions of UTIs. The manuscript was well written and easy to read. Some of the findings were expected. But reading the quotes did add some depth to the manuscript and my impression of how to care for patients with neurogenic bladder. I only have a few comments regarding improving the manuscript.

Comments:

1) Introduction line 75. I think 34% of patients with NB have AT LEAST one UTI diagnosed each year.

2) The methods section was clear and easy to follow. All of the analyses looked appropriate.

3) I think the themes about caregivers advocating for patients is good. I think a lot of the issues are related to the interplay between healthcare provider, patient, and caregiver when it comes to UTI symptoms. Some of the quotes in table 2 address this, but I was curious to see if the caregivers were ordering or advocating for patients to get UAs when the patients had non-specific symptoms. Do you have a sense of how much that is happening? Some of this is discussed in the negative perceptions section but it does not address some of the more common ASB complaints like malodorous or cloudy urine.

4) A lot of the discussion is a further description of the results. In your opinion, is there anything that readers should do differently when managing patients with SCI and neurogenic bladder with recurrent UTIs? I really felt like I didn't really understand how to translate these findings into clear next steps or take home points that would change practice.

Reviewer #2: This study addresses an important topic, to better understand individuals with neurogenic bladder’s perceptions about treatment for urinary tract infections. It is a qualitative study that uses focus groups of individuals with neurogenic bladder as well as some caregivers and thematic analysis to identify key concepts related to individuals with neurogenic bladder and urinary tract infections. They describe important topics for further research and intervention to improve the treatment of UTIs in patients who disproportionally suffer from them.

Overall Comments:

- More details about the choice of conceptual model and how that informed the development of the focus group guide and thematic analysis as well as description of Figure 2 would be useful

- A more explicit discussion of the inclusion of caregivers, if they were recruited to participate and if there were any limitations/benefits of caregiver participation may be worth including

Introduction

- A brief introduction to the conceptual framework used (the health belief model) may be beneficial to include in the introduction if there is room

Methods

- Was a qualitative reporting method used (SRQR?) if so this should be included

- The authors should justify the use of focus groups as the modality used in this study and why this was chosen over other modalities for gathering patient information (individual interviews, surveys, etc)

- There is mention of the participation of caregivers in the 3 of the 4 focus groups – what was their role ? Mention that their contributions were included when they were part of the focus groups.

- If there is room, specific mention of what iterative changes were made throughout the study may be helpful for readers

- A discussion of how many focus groups were performed and analyzed to determine saturation may be useful

Results

- In the demographics table, if median (range) number of UTIs among participants were obtained, this might help the reader understand the burden of UTIs among the participants

- Theme 2 – There is a lot of useful information then can be broken further into the specific provider, specialty setting, and then when they are seen in an unfamiliar setting

Discussion

- Line 345 – change medially to medically

- If there is space, could consider including your interpretations of the positive perceptions participants had about specialty units in treating UTIs

- Line 391 – change as to has

Figures

- I may be missing them but there are no titles to the figures

- Figure 2 needs some more explanation – either in the introduction or discussion

6. PLOS authors have the option to publish the peer review history of their article (what does this mean?). If published, this will include your full peer review and any attached files.

Reviewer #1: **Yes: **Michael Durkin

Reviewer #2: No

---

## [Author Response · Author response to Decision Letter 0]

27 Sep 2023

Reviewer #1: 

This is a qualitative analysis of patients perceptions of UTI in veterans with neurogenic bladder. This is a difficult patient population to treat. The team did a series of focus groups to understanding patients perceptions of UTIs. The manuscript was well written and easy to read. Some of the findings were expected. But reading the quotes did add some depth to the manuscript and my impression of how to care for patients with neurogenic bladder. I only have a few comments regarding improving the manuscript.

1) Introduction line 75. I think 34% of patients with NB have AT LEAST one UTI diagnosed each year.

Response:

We thank the reviewer for picking up on this detail. That is correct, we have made the edit to the Introduction.

2) The methods section was clear and easy to follow. All of the analyses looked appropriate.

Response:

We appreciate the commentary. 

3) I think the themes about caregivers advocating for patients is good. I think a lot of the issues are related to the interplay between healthcare provider, patient, and caregiver when it comes to UTI symptoms. Some of the quotes in table 2 address this, but I was curious to see if the caregivers were ordering or advocating for patients to get UAs when the patients had non-specific symptoms. Do you have a sense of how much that is happening? Some of this is discussed in the negative perceptions section but it does not address some of the more common ASB complaints like malodorous or cloudy urine.

Response:

We agree with the reviewer that caregiver influences are a key component of the complex process of UTI diagnosis, treatment and prevention in patients with NB. We did not actively recruit caregivers for our FGDs, so we were limited in assessing how caregiver influences may create or reinforce patient misconceptions about benign urine changes or other non-specific symptoms indicating UTI. Despite these limitations, several of our FGD participants described experiences where visiting nurses advocated for urine testing or other work-up/evaluation when the patient had appropriate signs or symptoms of possible UTI. For example: Table 2 quote B (bladder pain, bladder being ‘squeezed’) and quote D (fever). However, some participants described situations where the appropriateness of the signs or symptoms prompting a nurse to send urine testing was less clear (e.g., Table 4, quote G). In one of the FGDs where a patient’s wife participated, she indicated that malodorous and cloudy urine was often a prompt for her to suspect UTI. We have added this quote to Table 2 (now quote F) and the Results section (lines 258-260).

4) A lot of the discussion is a further description of the results. In your opinion, is there anything that readers should do differently when managing patients with SCI and neurogenic bladder with recurrent UTIs? I really felt like I didn't really understand how to translate these findings into clear next steps or take home points that would change practice.

Response:

We appreciate the thoughtful commentary, and we agree that there are substantial challenges to appropriately managing ASB and UTI in patients with NB. The overall aim and objective of this study was not to identify specific best practices (or specific methods to change existing practices) for clinical diagnosis and treatment of patients with NB and suspected UTI, but rather to understand patient experiences of ASB and UTI and examine how these experiences contribute to pervasive perceptions and beliefs that may impact diagnosis and treatment. We do offer some recommendations for how these findings can shape the components and structure of future antimicrobial stewardship interventions for patients with NB and suspected UTI (Discussion, lines 381-385 and lines 438-441).

Reviewer #2:

This study addresses an important topic, to better understand individuals with neurogenic bladder’s perceptions about treatment for urinary tract infections. It is a qualitative study that uses focus groups of individuals with neurogenic bladder as well as some caregivers and thematic analysis to identify key concepts related to individuals with neurogenic bladder and urinary tract infections. They describe important topics for further research and intervention to improve the treatment of UTIs in patients who disproportionally suffer from them.

Overall Comments:

- More details about the choice of conceptual model and how that informed the development of the focus group guide and thematic analysis as well as description of Figure 2 would be useful

Response:

We have added more information about the choice of the HBM and how that informed data collection and analysis to the Methods (lines 162-174).

- A more explicit discussion of the inclusion of caregivers, if they were recruited to participate and if there were any limitations/benefits of caregiver participation may be worth including

Response:

We have added information about inclusion of caregivers to the Methods (lines 148-152).

Introduction

- A brief introduction to the conceptual framework used (the health belief model) may be beneficial to include in the introduction if there is room

Response:

In the interest of space and so as not to be redundant, we have not included information about the HBM in the Introduction. However, we have added significant information about the HBM and how it was used in this study to the Methods (see response above).

Methods

- Was a qualitative reporting method used (SRQR?) if so this should be included

Response: We used the COREQ checklist and have added that to the Methods section.

- The authors should justify the use of focus groups as the modality used in this study and why this was chosen over other modalities for gathering patient information (individual interviews, surveys, etc)

Response: We have added this information to the Methods section (lines 120-122).

- There is mention of the participation of caregivers in the 3 of the 4 focus groups – what was their role ? Mention that their contributions were included when they were part of the focus groups.

Response: 

As discussed above, we have added information about caregiver involvement to the Methods section. Caregivers were not specifically recruited, but their participation was not prohibited. A total of 4 non-medically trained caregivers (spouses/partners) participated in 3 different FGDs. 

- If there is room, specific mention of what iterative changes were made throughout the study may be helpful for readers

Response:

Iterative changes were mostly minor modifications to the question prompts and follow-up probes in the focus group discussion guide. For example, slightly modifying a question or probe in response to a participant being confused about what that question was asking. We also made minor modification to follow-up probes based on emerging concepts and themes from the initial discussions. For example, in our first FGD, a participant described a specific regimen of urinary catheter care and maintenance he had developed with his primary care physician for UTI prevention. This led us to modify the follow-up probes for a question on knowledge of UTI prevention methods. We have added information about this to the Methods section (Lines 173-176).

- A discussion of how many focus groups were performed and analyzed to determine saturation may be useful

Response:

We have added additional information about the determination of the number of focus groups performed and saturation to the Methods section (lines 198-203).

Results

- In the demographics table, if median (range) number of UTIs among participants were obtained, this might help the reader understand the burden of UTIs among the participants

Response:

We collected data on UTI frequency of our focus group participants based on the administrative (ICD10) diagnosis codes for UTI between 2017-2018 at our study sites. We have included this information in the Methods section (lines 144-147). We have also added it to Table 1 and the Results section (lines 210-211). 

- Theme 2 – There is a lot of useful information then can be broken further into the specific provider, specialty setting, and then when they are seen in an unfamiliar setting

Response:

We agree with the reviewer that there is substantial helpful information in the data supporting Theme 2. However, we feel that the combination of the text in the Results section and the quotes in the Table thoroughly represent the key findings from this theme. In keeping with the journal’s goals for authors to present and discuss their findings concisely, and considering the manuscript’s already sizeable length, we would be happy to provide additional information to individual readers upon request if needed. 

Discussion

- Line 345 – change medially to medically

Response: We have made this change. 

- If there is space, could consider including your interpretations of the positive perceptions participants had about specialty units in treating UTIs

Response: We have added the specialty SCI/D units into our Discussion of positive perceptions of UTI care (lines 402-410).

- Line 391 – change as to has

Response: We have made this change.

Figures

- I may be missing them but there are no titles to the figures

Response: 

As is indicated in the PLOS One Guidelines for Authors section on Figures, we have included titles in the Figure captions in the manuscript file itself. We do not see a requirement for the Figure files themselves to have titles as well, so we defer to the Editor’s suggestion on this critique.

- Figure 2 needs some more explanation – either in the introduction or discussion

Response: 

As per our response above, we have added additional information about the HBM and its constructs to the Methods section.

---

## [Editor Report · Decision Letter 1]

19 Oct 2023

Perceptions, experiences, and beliefs regarding urinary tract infections in patients with neurogenic bladder: a qualitative study

PONE-D-23-15723R1

Dear Dr. Fitzpatrick,

We’re pleased to inform you that your manuscript has been judged scientifically suitable for publication and will be formally accepted for publication once it meets all outstanding technical requirements.

Kind regards,

Maria Berghs, PhD

Academic Editor

PLOS ONE
---

## [Editor Report · Acceptance letter]

23 Oct 2023

PONE-D-23-15723R1 

Perceptions, experiences, and beliefs regarding urinary tract infections in patients with neurogenic bladder: a qualitative study 

Dear Dr. Fitzpatrick:

I'm pleased to inform you that your manuscript has been deemed suitable for publication in PLOS ONE. Congratulations! Your manuscript is now with our production department. 

Kind regards, 

on behalf of

Dr. Maria Berghs 

Academic Editor

PLOS ONE